# Influence of Sintering Atmosphere, Temperature and the Solution-Annealing Treatment on the Properties of Precipitation-Hardening Sintered 17-4 PH Stainless Steel

**DOI:** 10.3390/ma16020760

**Published:** 2023-01-12

**Authors:** Jan Kazior

**Affiliations:** Department of Material Engineering, Faculty of Materials Engineering and Physics, Cracow University of Technology, 31-155 Krakow, Poland; jan.kazior@pk.edu.pl

**Keywords:** powder metallurgy, stainless steel, sintering, atmosphere, dilatometry

## Abstract

So far, unlike metal injection molding (MIM), conventional powder metallurgy technology (PM) has not been regarded as a method for producing structural elements from 17-4 PH powders, due to the problems of obtaining almost fully compacted shapes after sintering. Nevertheless, recent research demonstrates that it is possible to manufacture sintered parts with high strength by pressing and sintering. The purpose of the study was to determine the degree of densification of 17-4 PH sintered stainless steel during sintering at different temperatures and atmospheres. As a result of the study, it was pointed out that both the temperature and the sintering atmosphere play an essential role in the process of densification of the studied powders during sintering. The formation of delta ferrite and a more pronounced degree of spheroidization of the pores is activated by a higher sintering temperature. Furthermore, after solution-annealed and age-hardened treatment, sintered 17-4 PH stainless steel exhibits high strength with moderate ductility at a level that is difficult to achieve for other sintered stainless-steel grades, such as austenitic, ferritic and martensitic. In turn, the largest improvement in the pitting corrosion resistance in 0.5 M NaCl solution is reached by sintering at 1340 °C in hydrogen and after solid solution treatment.

## 1. Introduction

Stainless steels can be classified into several groups. These include: austenitic, ferritic, martensitic, precipitation hardening, dual-phase and duplex stainless steels. There are a lot of commercially available stainless-steel powder grades. A wide range of them can be produced by the water atomization process and then pressed and sintered. Precipitation-hardening stainless-steel powder is a relatively new family of alloys. These are designated to provide high strength and toughness by the submicroscopic precipitates in the matrix [1]. Most of the published research papers on the manufacture of sintered 17-4 stainless steel preferred injection molding technology, while there are few publications devoted to the fabrication of parts by cost-effective energy and material-saving powder metallurgy technology. However, recent published research works most clearly indicate that it is possible to develop high-strength sintered parts by the traditional pressing and sintering technique [2,3,4,5,6,7]. The addition of such elements as copper and niobium, which form intermetallic precipitates during aging, cause the strengthening of 17-4 PH stainless steels. The presence of copper promotes the precipitation-hardening process, while niobium contributes to reductions in hardness after solution annealing, thus, making machining possible and hindering over aging.

To make it easier to complete phase transformation, the nickel and molybdenum contents are limited. Water-atomized 17-4 PH stainless powders have martensitic structures. During solution annealing, the matrix forms with precipitates, forming elements of supersaturated austenite solid solution, which is transformed into martensite during cooling. Upon aging, second-phase precipitates nucleate uniformly throughout the matrix. Aging treatment is designed to ensure that the precipitates nucleate uniformly throughout the matrix, limiting the displacement of dislocations and, thus, causing an increase in the hardness and strength of sintered steel.

In the last study, it was confirmed that high-strength sintered stainless steels, such as 17-4 PH, achieving full or near-full density is essential to realize the full benefit of their superior mechanical properties. It has been shown that in injection molding technology, a properly conducted debinding process and appropriate selection of other technological parameters, such as sintering temperature and sintering atmosphere, make it possible to obtain a high sintered density that will ensure high tensile strength in sintered steels. Further, the carbon remaining in the structure is crucial for the corrosion resistance and strength properties of the sintered steel. From the literature review, it is clear that only a few efforts have been undertaken to understand the densification and shrinkage mechanism during the sintering of pre-alloyed 17-4 PH stainless-steel powder [8,9].

The conducted research aimed to study the impact of technological parameters in the traditional process of pressing in rigid dies and sintering on the densification, mechanical properties and pitting corrosion resistance in 0.5 NaCl solution of the 17-4 PH sintered stainless steel, either in the as-sintered or heat-treated conditions.

## 2. Materials and Methods

Water-atomized 17-4 PH stainless-steel powder with the following chemical composition in % wt. (C-0.027; Si-0.73; Cr-16.28; Ni-4.28; Cu-4.04; Nb-0.32; Mn-0.05; P-0.015; Fe-balance) provided by Ametek (Berwyn, PA, USA) was used. The average particle size of the powders was 55 µm. The particle size distribution of the powder is shown in Figure 1, its apparent density is 2.54 g/cm^3^ and the flow rate 31 (s/50 g).

The compressibility of 17-4 PH powder was studied at 400–700 MPa compacting pressures for 20 × 5 mm^3^ cylindrical specimens. As reference materials, AISI 316 L and AISI 410 L powders were also investigated. The dilatometer bar specimens 5 × 5 × 15 mm^3^, tensile specimens for mechanical features and cylindrical specimens of size ∅20 × 5 (mm^3^) for precipitation-hardening treatment and for corrosion behavior were uniaxially compacted in rigid die at 600 MPa. All compacts were thermally debound at 450 °C for 40 min in pure dry hydrogen before sintering.

Dilatometric analysis was carried out in a horizontal NETZSCH 402 PC dilatometer (Selb, Germany) under pure dry hydrogen, vacuum and nitrogen/hydrogen atmosphere at two isothermal sintering temperatures of 1240 °C and 1340 °C for 60 min and 120 min isothermal holding. The thermal cycle was heating to isothermal sintering temperatures at rates of: 1, 5, 10 and 20 °C/min, followed by cooling at a rate of 20 °C/min.

Sintering for tensile bar specimens and cylindrical specimens were carried out in Nabertherm^®^ P 330 (Lilienthal, Germany) tube furnace at 1340 °C for 60 min. Then, some specimens were subjected to a solution-annealing treatment at 1040 °C for 60 min in a hydrogen atmosphere. Then, some of the tested samples after solution-annealing treatment were subjected to an aging treatment at 480, 490 and 500 °C in a nitrogen atmosphere.

The Archimedes method was adopted to measure sintered density in the samples. Metallographic characterization was carried out via light optical microscope (LOM) on specimens polished and etched employing standard metallographic procedures. The hardness (HV) was determined.

Tensile tests were carried out on a standard tensile machine at a crosshead speed of 1 mm/min, in accordance with the ISO 3928 test method.

The corrosion distinctive of the sintered stainless steels was tested using ATLAS 0531 EU&IA (ATLAS—SOLLICH) (Rębiechowo k/Gdańska, Poland) including reference electrode, a counter electrode and a working electrode. The reference electrode was a saturated calomel electrode (SCE) and counter electrode was a platinum electrode. The specimen was a working electrode. The testing environment was a 0.5 M NaCl solution at ambient temperature. The potentiodynamic polarization and open-circuit potential (OCP) measurement tests were performed. Before corrosion test, the samples were degreased, cleaned in distilled water and next in acetone and dried. The corrosion check began with the OCP and the potential of the samples was registered and tracked as a function of time until it obtained a steady value. After the OCP measurement, the potentiodynamic test was performed at a rate of 1 mV/s, beginning from 200 mV below the OCP up to 1 V.

## 3. Results

The compressibility of 17-4 PH powder concerning conventional stainless-steel powders AISI 316 L and AISI 410 L is shown in Figure 2. In principle, the green density of 17-4 PH stainless steel is lower regarding other highly alloyed stainless steels; however, the investigated powders exhibit reasonable compressibility to process them by conventional compaction and sintering.

For powder metallurgy technology and, in particular, in the case of stainless steels, the most important parameters in the lubricant-removal process are the heating rate, the temperature of the isothermal sintering and the time of isothermal holding. These conditions are intended to ensure gradual, rather than rapid, decomposition of the lubricant and its complete removal. Figure 3 provides an example of the TG curve for 17-4 PH powder with a heating rate of 10 K/min in an argon atmosphere. At the same time, as a result of tests in other atmospheres, it was further noted, in addition to the previously mentioned parameters, that the type of gas used also affects the removal temperature of the lubricant. The decrease in the removal temperature of a lubricant when helium or hydrogen is used is related to the higher thermal conductivity of these gases compared to argon or nitrogen, for example, which results in faster heating of the green compacts. In addition, helium atoms or hydrogen molecules, due to their size, are much smaller compared to argon atoms or nitrogen molecules, which causes them to enter the structures of the porous material more quickly and transfer heat more quickly to the lubricant as well as the green compacts. For manufacturers of sintered products made from stainless-steel powders, this phenomenon is very important to ensure the gradual and total removal of a lubricant.

Furthermore, as expected, the lower green density of 17-4 PH compacts transforms into lower sintered density. As can be seen from Figure 4, the density of compacts sintered at 1240 °C is influenced not only by the green density but also by the sintering environment—in a hydrogen atmosphere, higher densification occurs than in a vacuum [10].

In Figure 5, the dilatometric dimensional changes during the sintering of 17-4 PH stainless steel compacts at 1240 °C and 1340 °C for 60 min in hydrogen and a vacuum are presented. As can be observed from the course of dimensional changes up to 900 °C, thermal expansion prevails and then above 900 °C, contraction begins, which indicates the beginning of mass transport phenomena. From the evaluation of dilatometric curves, it can be deduced that shrinkage is influenced both by the sintering temperature and the sintering atmosphere. With increasing sintering temperature, shrinkage increased for both sintering atmospheres; however, the higher shrinkage is observed for hydrogen as compared with a vacuum, in particular, for lower sintering temperatures. Further, metallographic studies of sintered 17-4 PH stainless steels in the as-unetched state indicate that as the sintering temperature increases, both in hydrogen and vacuum, a clear increase in the degree of the spheroidization of pores can be observed, as can be seen in Figure 6 and Figure 7.

Detailed analysis of dimensional changes indicates that the shrinkage rate for sintering at lower temperatures is roughly stable during heating above 900 °C and isothermal holding. On the contrary, for higher sintering temperatures, linear shrinkage is significantly higher but the shrinkage rate starts to decrease during isothermal holding. During cooling from sintering temperature, both in hydrogen and vacuum, at temperatures near to 200 °C, expansion can be observed, which is the result of the transformation of austenite into martensite.

For a better understanding of the dimensional behaviour of the study material, the shrinkage rate versus temperature during heating up to 1340 °C for hydrogen and in vacuum are presented in Figure 8. During heating, three distinct peaks of the shrinkage rate can be observed. At a temperature near 750 °C, the first peak of shrinkage rate is associated with the bct martensite transformation to γ austenite. In a temperature range 1050–1200 °C, a second peak can be observed and can be attributed to the offset of thermal expansion by initial sintering shrinkage. The third-most visible is the result of activated sintering, initiated by a sudden shrinkage beginning near 1250 °C and may be related to the fracture of silica [11], which covers the powder particles and, thereby, causes activated sintering in the solid state associated with probable initial particle rearrangement.

Analysis of the microstructure indicates that during cooling, the transformation of austenite into a martensitic structure takes place. Thus, after sintering in vacuum and hydrogen, δ ferrite and martensite can be distinguished in the structure. Since the files from martensite and δ ferrite overlap in XRD studies, it was decided to carry out additional sintering in a nitrogen–hydrogen atmosphere to confirm that the δ ferrite affects the densification of the compact during sintering. The analysis of the test results from the thermal analysis correlates well with the XRD results shown in Figure 9. On the other hand, in Figure 10, thermodynamic calculations for the 17-4 PH sintered stainless steel are presented, as a result of which a pseudo-double plot was developed.

To better understand the role of the atmosphere, an additional sintering in a 95%N_2_/5%H_2_ gas mixture was performed. The respective dimensional changes in comparison with hydrogen- and vacuum-sintered compacts are presented in Figure 11. It is evident that sintering in a N_2_/H_2_ atmosphere gives incomplete densifications due to the stabilization of austenite by diffusion of nitrogen and, in consequence, the transformation of austenite to δ ferrite does not occur.

In addition, the longer isothermal sintering time and different heating rate were examined to study the densification behaviour of the 17-4 PH stainless-steel compact. From Figure 12, it is seen that prolonged isothermal sintering time from 60 to 120 min gives slightly higher densification and the density of dilatometric samples increased from 6.95 g/cm^3^ to 7.12 g/cm^3^, respectively. As can be seen in Figure 13 and Figure 14, the increasing heating rate gives lower shrinkage and, for example, for heating rates 1 °C/min and 20 °C/min, the sintered density of dilatometric samples is 7.16 g/cm^3^ to 7.09 g/cm^3^, respectively.

Precipitation-hardened stainless steels are subjected to heat treatments consisting of solution annealing and aging to improve mechanical properties. Sintered 17-4 PH stainless steels were subjected to solution-annealing treatments at temperatures of 1020–1040 °C and then aged in a temperature range of 480–500 °C. The results of HV hardness measurements are shown in Figure 15, while the mechanical properties of the tensile test are shown in Table 1.

As a result of the study of the strength properties of sintered 17-4 PH stainless steels, the results obtained show that it is possible to attain a tensile strength after solution annealing (1040 °C) and aging of 1147 MPa and an elongation of 2.4%. If it is necessary to increase plastic properties, the aging temperature should be increased to 550 °C, obtaining an elongation of 3–4%. Increasing the plastic property results in decreasing the strength properties to a level of 800–1000 MPa.

An example of the microstructure of the sintered 17-4 PH stainless steels in the non-etched state is shown in Figure 16. The matrix of the sintered steel is martensite with a small amount of delta ferrite and visible rounded pores. The presence of delta ferrite during high-temperature sintering certainly promotes the spheroidization of pores and extensive densification of the sintered material. In addition, the dispersive precipitates, as a result of the heat treatment, significantly improve the mechanical properties. However, the fine dispersive precipitates are not visible under an optical microscope.

In addition, fracture findings on the SEM well correlated with the results of tensile tests. After sintering (Figure 17) and solution-annealed treatment (Figure 18), the fracture surface is ductile. In contrast, after aging (Figure 19), the fracture surface indicates some brittle areas.

Stainless steels are significantly resistant to general corrosion, but in aggressive environments (in particular, those containing chlorides), they are prone to various forms of localized corrosion (pitting, crevice, intergranular, stress corrosion cracking). Pitting is recognized as the most dangerous type of corrosion because it is very difficult to detect and also to ensure adequate protection. Pitting is the most common type of corrosion in stainless steel. It is manifested in the form of small pits on passive metal surfaces.

Open-circuit potential (OCP) changes were measured for all test steels immersed in 0.5 m NaCl solution and the results are shown in Figure 20. The description of a sample designation applied in the following part of this article is given in Table 2.

The potential of sintered steel shows the tendency to slightly reduce with time. After 30 min of exposure in 0.5 M NaCl solution, the sample almost reaches a steady state and OCP potential is about −556 mV. The potential of solution-annealed steel is more positive and equals −380 mV. From the analysis of the presented characteristic, it can be concluded that solution-annealing treatment leads to a potential increase (shift to more positive values) in comparison to the sintering process while, after aging, the treatment potential of 17-4 PH sintered steels is reduced.

Figure 21 shows the polarization curves of the tested sintered 17-4 PH stainless steels. As expected, sintered 17-4 PH stainless steel does not show typical anodic polarization curves consisting of an active, passive and transpassive region. The typical maximum of the active–passive transition does not appear. There is a rapid increase in current density and destruction of the passive layer and transition to the pitting corrosion area.

In the case of the sample after solution-annealed treatment, the polarization curve is different. An active–passive transition maximum and an active, passive and transpassive region may be observed. Similar polarization curves were obtained for steel aging at a temperature higher than 480 °C.

Results of the performed electrochemical test indicate that by applying a solution-annealing treatment after the sintering process, resistance to pitting corrosion slightly increases. Furthermore, the potentiodynamic polarization measurements reveal that solution-annealing and aging treatment at 480 °C leads to optimum corrosion resistance in a 0.5 M NaCl solution, including higher OCP, polarization resistance and pitting potential. On the other hand, aging at 500 °C results in the deterioration of corrosion resistance.

## 4. Conclusions

Sintering temperature and atmosphere play a significant role in the densification during the sintering of 17-4 PH stainless-steel compact. The increase in sintering temperature results in an increase in density, which is related to the fact that a higher sintering temperature promotes the formation of delta ferrite, which affects the degree of pore spheroidization and an increase in the degree of densification. In contrast, a longer isothermal sintering time and higher heating rate slightly increased the sintered density of samples. Furthermore, as expected, higher densification occurs in hydrogen than in a vacuum. Moreover, the sintering in a N_2_/H_2_ atmosphere gives incomplete densifications since the nitrogen in the solid solution prevents the formation of delta ferrite, which enhances the sintering and densification.

After the solution-annealing and ageing treatment, the precipitating hardening sintered 17-4 PH stainless steel shows high strength with reasonable ductility levels that are hardly achievable for austenitic, ferritic and martensitic sintered stainless-steel grades.

The largest improvement in corrosion resistance of sintered 17-4 PH stainless steel is achieved by sintering at 1340 °C in hydrogen after a solid solution-annealing treatment. On the other hand, after solution annealing at 480 °C, the resistance to pitting corrosion in a 0.5 NaCl solution (higher OCP, polarization resistance and pitting potential) is the highest, as compared to solution annealing at 500 °C, where the corrosion resistance decreases significantly.

## Figures and Tables

**Figure 1 materials-16-00760-f001:**
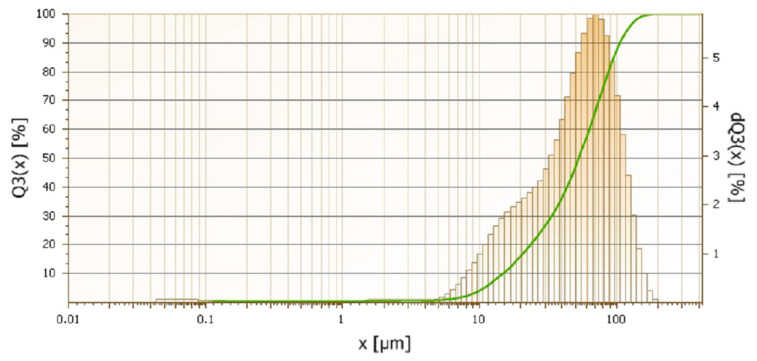
Laser measurements of particle size distribution of water-atomized 17-4 PH stainless-steel powder.

**Figure 2 materials-16-00760-f002:**
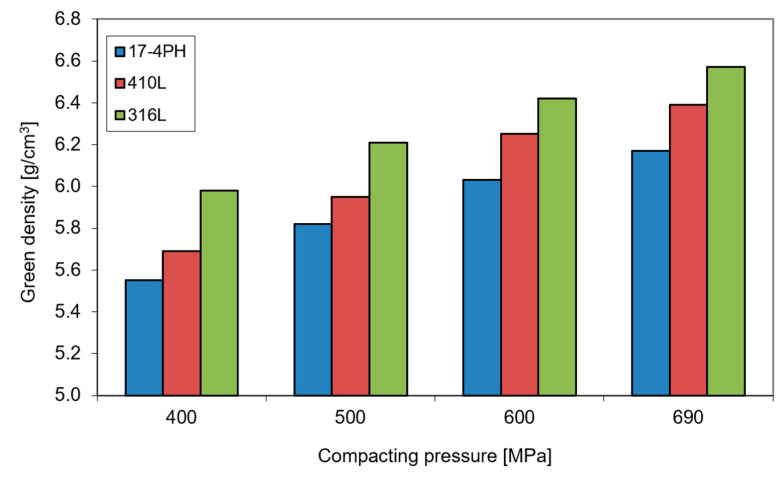
The green density of 17-4 PH, AISI 410 L and AISI 316 L stainless-steel compacts as a function of compacting pressure.

**Figure 3 materials-16-00760-f003:**
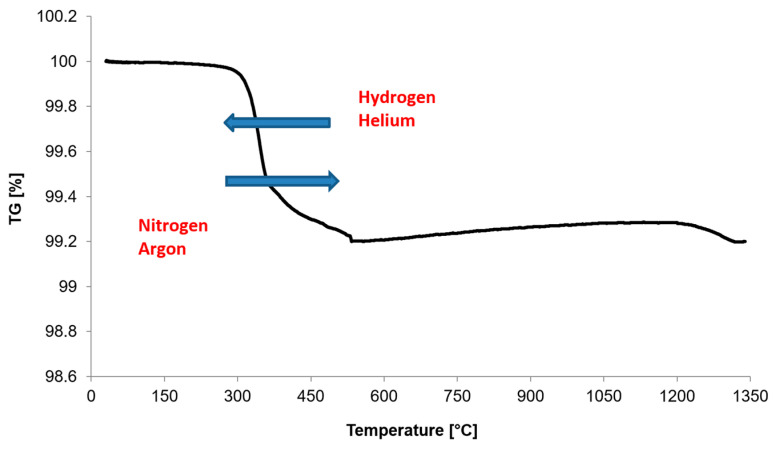
TG curve of 17-4 PH powders during heating to isothermal sintering temperature at 1340 °C.

**Figure 4 materials-16-00760-f004:**
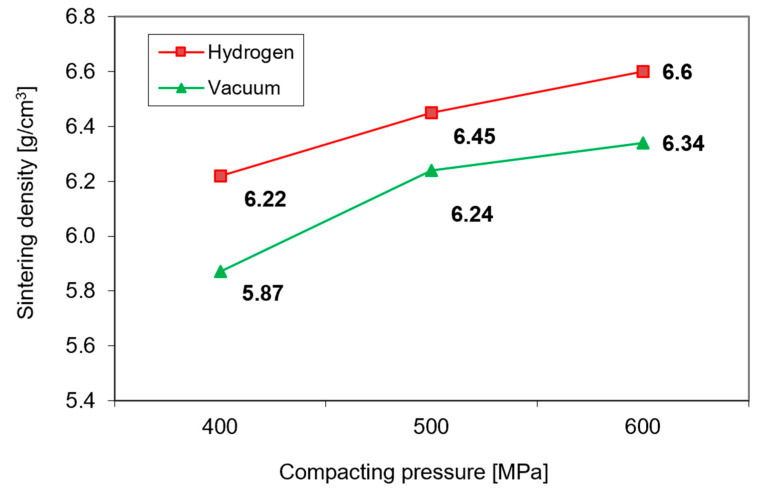
The effect of compacting pressure on the sintered density of 17-4 PH stainless steel after sintering at 1240 °C in hydrogen and vacuum.

**Figure 5 materials-16-00760-f005:**
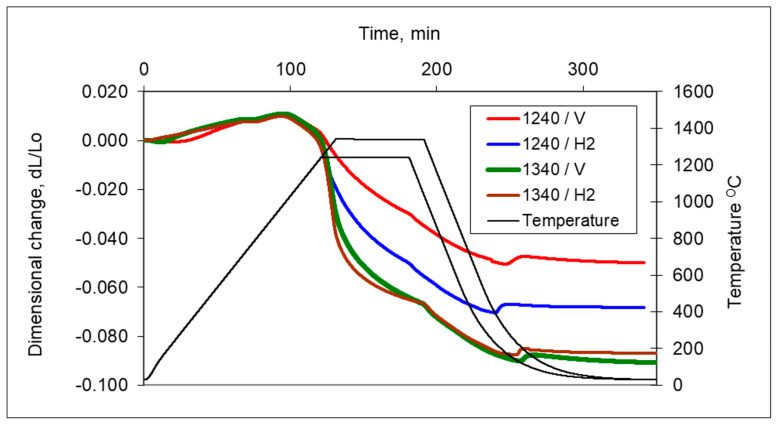
Dimensional changes in 17-4 PH stainless-steel compact during heating to sintering at 1240 °C and 1340 °C for 60 min in hydrogen and a vacuum.

**Figure 6 materials-16-00760-f006:**
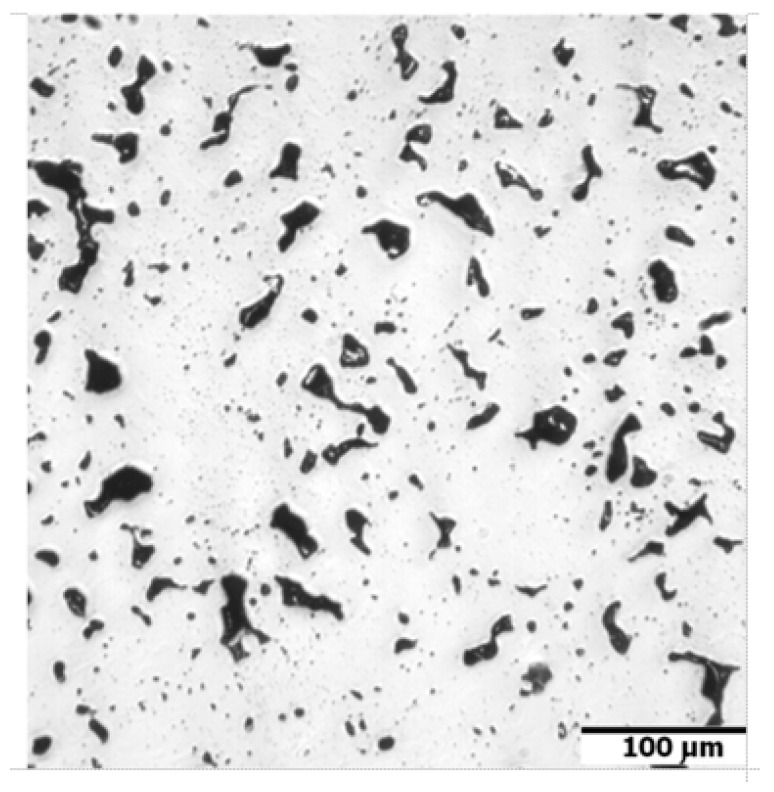
Microstructure of sintered 17-4 PH stainless steel after sintering at 1240 °C for 60 min in pure dry hydrogen, no etched.

**Figure 7 materials-16-00760-f007:**
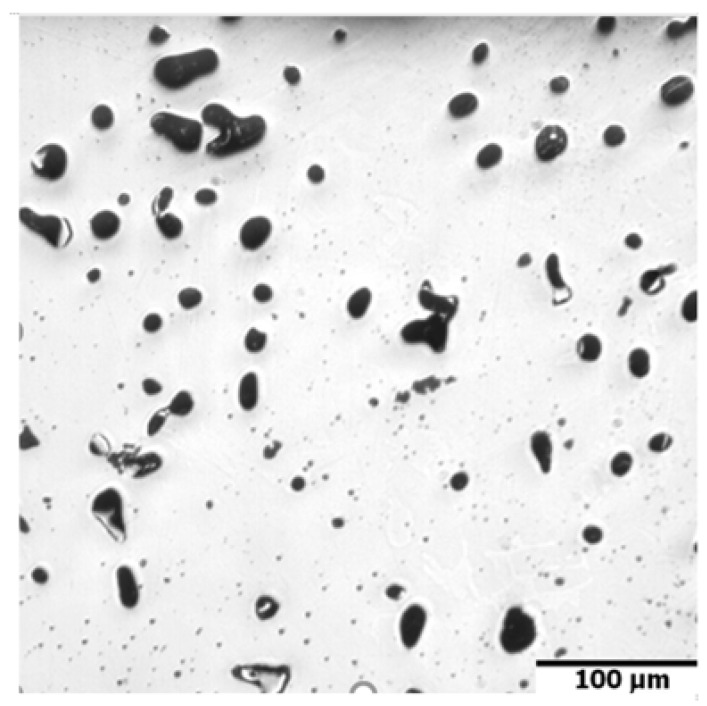
Microstructure of sintered 17-4 PH stainless steel after sintering at 1340 °C for 60 min in pure dry hydrogen atmosphere, no etched.

**Figure 8 materials-16-00760-f008:**
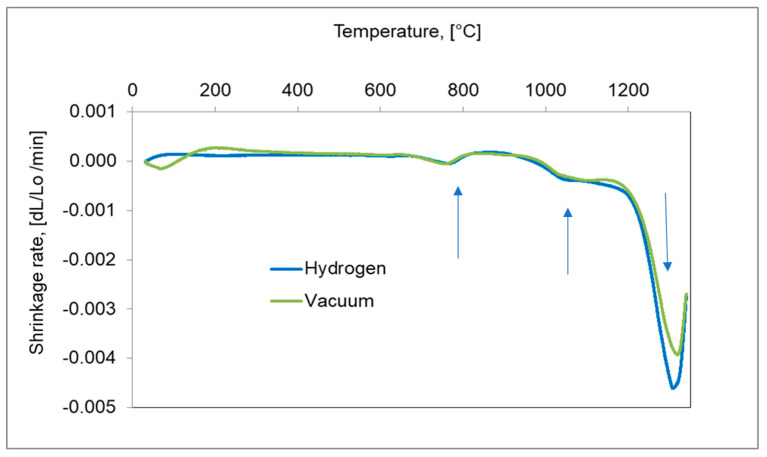
Shrinkage rate of 17-4 PH stainless-steel compact during heating to 1340 °C with the heating rate at 10 °C/min in hydrogen and a vacuum.

**Figure 9 materials-16-00760-f009:**
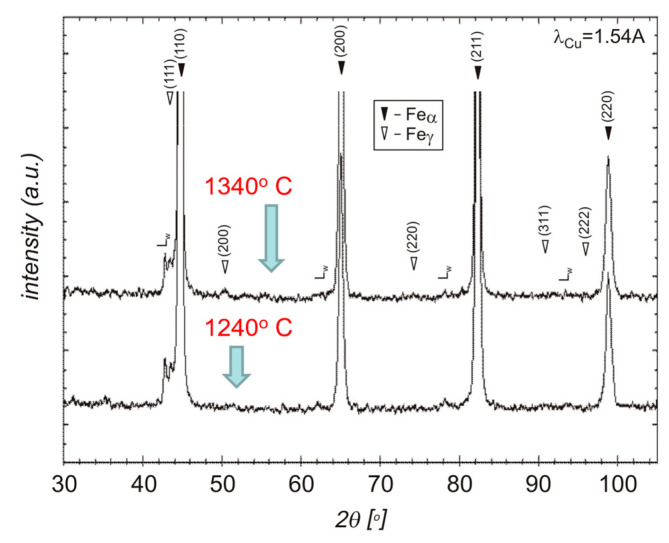
XRD analysis of sintered 17-4 PH stainless steel.

**Figure 10 materials-16-00760-f010:**
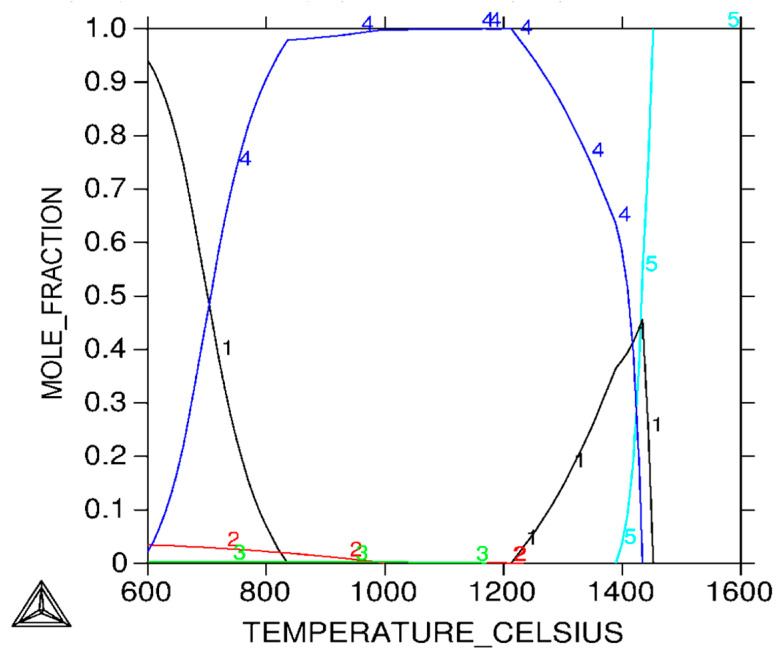
Thermocalc diagram for 17-4 PH powder, 1. ferrite (δ), 2. copper FCC precipitation, 3. niobium Fcc precipitation, 4. austenite (γ), 5. liquid (5).

**Figure 11 materials-16-00760-f011:**
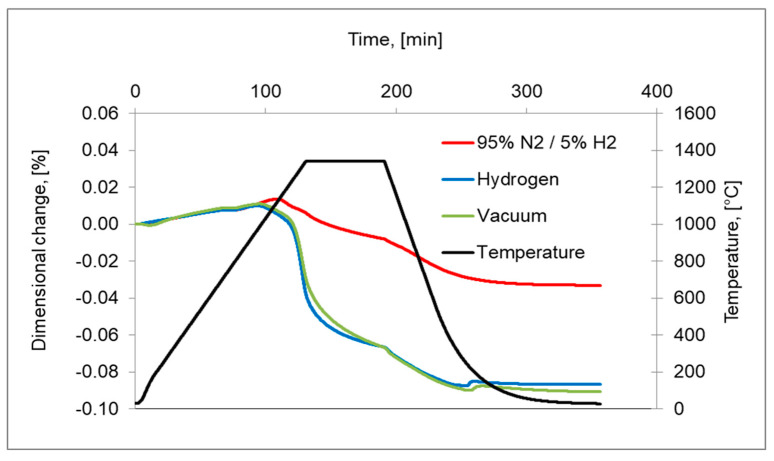
Dilatometric curves of 17-4 PH sintered stainless-steel compacts in hydrogen, vacuum, 95%N_2_-5%H_2_ in temperature 1340 °C with 10 °C/min heating rate and for 60 min isothermal holding.

**Figure 12 materials-16-00760-f012:**
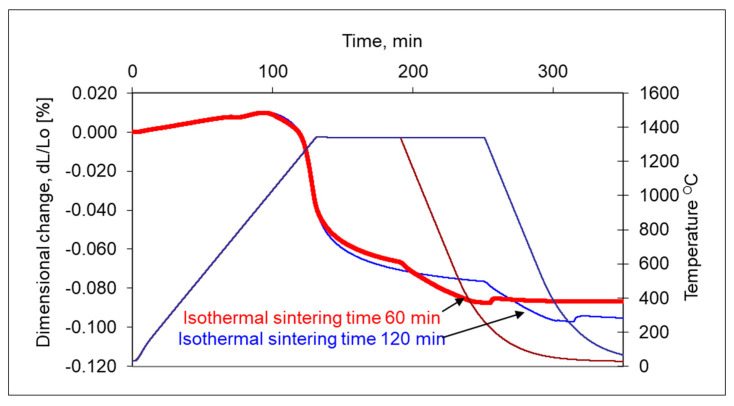
Influence of isothermal holding time for shrinkage of 17-4 PH stainless-steel compacts when sintering at 1340 °C with the heating rate at 10 °C/min in hydrogen for 60 min and 120 min isothermal holding.

**Figure 13 materials-16-00760-f013:**
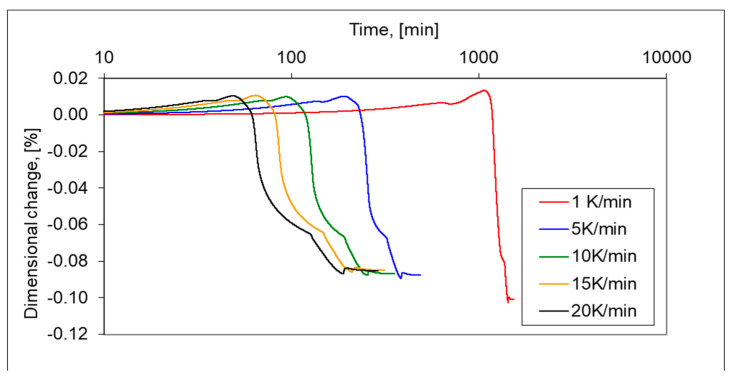
Dilatometric curves of 17-4 PH stainless-steel powders sintered in hydrogen, at 1, 5, 10, 15 and 20 °C/min heating rate for isothermal sintering at a temperature of 1340 °C.

**Figure 14 materials-16-00760-f014:**
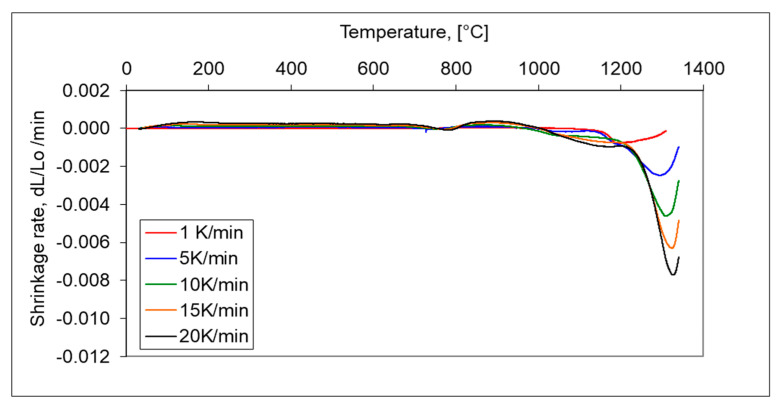
Shrinkage rate of 17-4 PH stainless-steel compacts sintered in hydrogen at 1, 5,10, 15 and 20 °C/min heating rate for isothermal sintering at a temperature of 1340 °C.

**Figure 15 materials-16-00760-f015:**
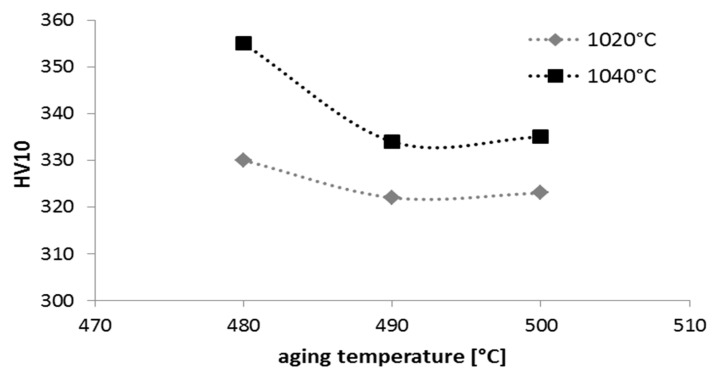
Hardness HV_10_ of sintered 17-4 PH stainless steel at 1340 °C for 60 min in hydrogen and then subjected to solution-annealed treatment at 1020–1040 °C and finally aged at 480–500 °C.

**Figure 16 materials-16-00760-f016:**
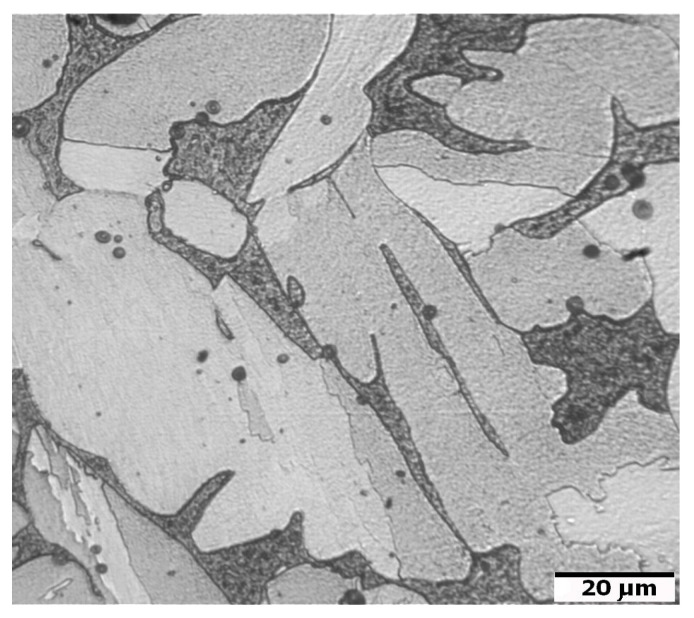
Microstructure of sintered 17-4 PH stainless steel at 1340 °C for 60 min in a pure dry hydrogen atmosphere.

**Figure 17 materials-16-00760-f017:**
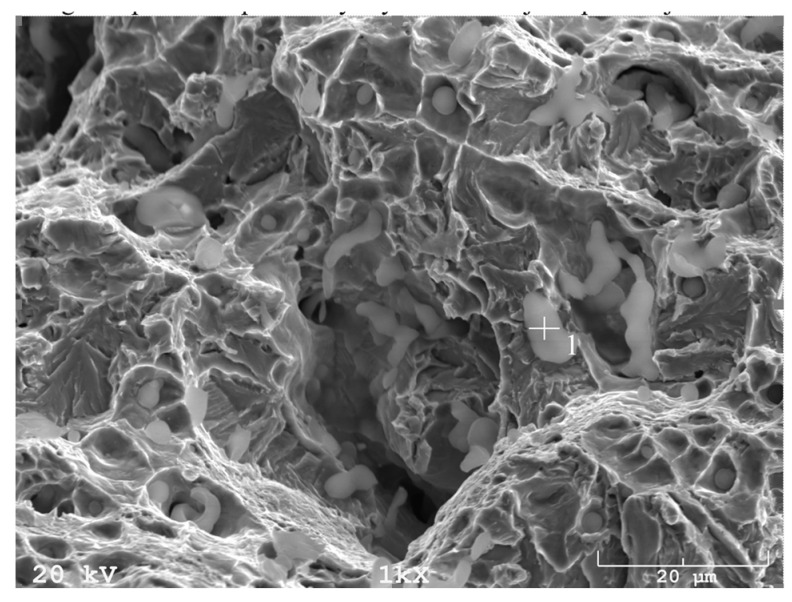
SEM fracture of sintered 17-4 PH stainless steel at 1340 °C for 60 min in hydrogen.

**Figure 18 materials-16-00760-f018:**
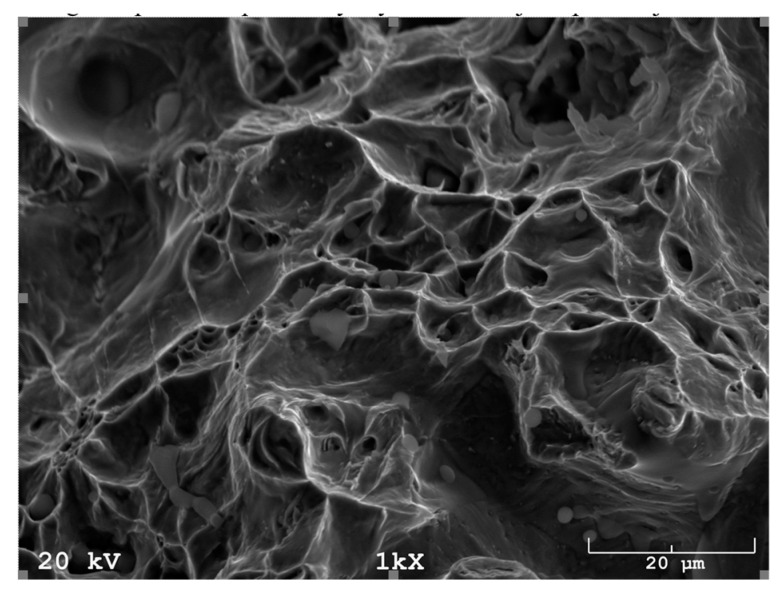
SEM fracture of sintered 17-4 PH stainless steel at 1340 °C for 60 min in hydrogen and subsequent solution annealing at 1040 °C.

**Figure 19 materials-16-00760-f019:**
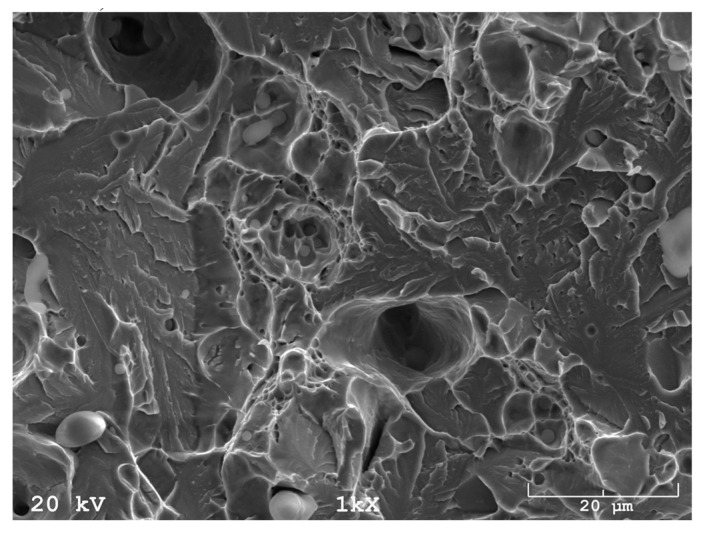
SEM fracture of sintered 17-4 PH stainless steel at 1340 °C for 60 min in hydrogen and subsequent solution annealing at 1040 °C and aging at 480 °C.

**Figure 20 materials-16-00760-f020:**
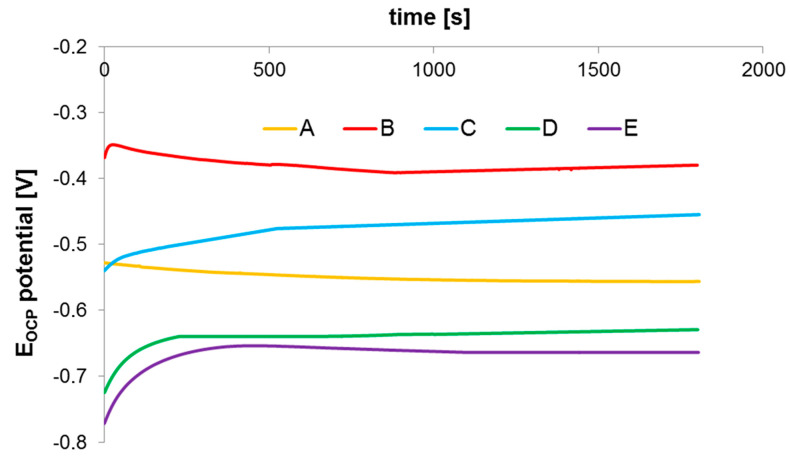
OCP changes after being immersed for 30 min in 0.5 M NaCl solution fot 17-5 PH sintered stainless steel.

**Figure 21 materials-16-00760-f021:**
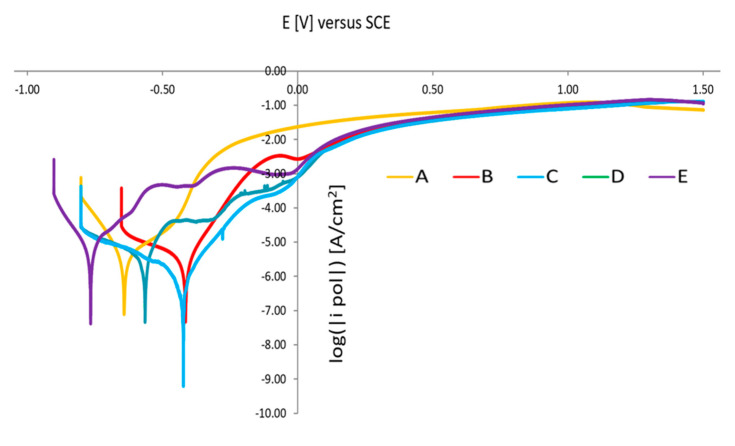
Potentiodynamic polarization curves of 17-4 PH sintered steels in 0.5 M NaCl solution.

**Table 1 materials-16-00760-t001:** Mechanical properties of sintered 17-4 PH stainless steel at 1340 °C and subsequent solution annealing at 1040 °C and aging at 480 °C.

Treatment	UTS {MPa]	Elongation [%]
Sintering	883	2.1
Sintering and solution annealing	743	1
Sintering, solution annealing and aging	1147	2.4

**Table 2 materials-16-00760-t002:** Description of investigated samples.

Designation of Sample	A	B	C	D	E
Condition of treatment	sintering	Sintering, Solution- annealing	sintering, solution-annealing, aging (480 °C)	sintering, solution-annealing, aging (490 °C)	sintering, solution-annealing, aging (500 °C)

## Data Availability

Data presented in this article are available at request from the author.

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
