# Peer review of "Influence of Sintering Atmosphere, Temperature and the Solution-Annealing Treatment on the Properties of Precipitation-Hardening Sintered 17-4 PH Stainless Steel"

_materials, 2023, doi:10.3390/ma16020760_

Round 1

Reviewer 1 Report

Comments:

1) Check spelling "atmized"   Line 45 , Page 1

2) Powder particle size considered for this research work may be mentioned.

3) Compaction dwell time may be mentioned in Line 81, page 3

4) Authors may mention the powder particle size used in this research work and also the grain refinement that has happened after ball milling operation. Moreover, the particle size may influence the hardness of the sample which has not been addressed.  

5) Authors may provide some more explanation on "the liquidus and solidus phase transformation" 

6) More recent references are to be added. (The current list is not sufficient and few seems to be old)

Author Response

Dear Reviewer

Manuscript ID: Materials - 2121817

I have attached my answer to your comments.

Yours sincerely

Jan Kazior

Reviewer 2 Report

Morphology, sintering microstructure and mechanical properties of 17-4 PH stainless steel prepared by traditional PM route were investigated in detail, the results were sound with well organized structure.

1. Besides the linear shrinkage of the samples sintered in different atmosphere, related microstructure should also be given to illustrate the difference caused by the reaction with the gas.

2. The strength of the steel after aging was the lowest, why?

3. what's the effect of heating rate on the sintering behavior?  Usually, a moderate heating rate was used during the fabrication.

Author Response

Dear Reviewer

Manuscript ID: Materials - 2121817

Please find attached my answer to your comments.

Yours sincerely

Jan Kazior
